Ac2-26 attenuates hepatic ischemia-reperfusion injury in mice via regulating IL-22/IL-22R1/STAT3 signaling

Li Wanzhen 1
Jiang Hongxin 2
Bai Chen 1
Yu Shuna 1
Pan Yitong 1
Wang Chenchen 1
Li Huiting 1
Li Ming 1
Sheng Yaxin 1
Chu Fangfang 1
Wang Jie 1
Chen Yuting 1
Li Jianguo ljg7111@wfmc.edu.cn 1
Jiang Jiying jiangjy@wfmc.edu.cn 1
1 Department of Anatomy, Weifang Medical University , Weifang , Shandong , China
2 Morphology Lab, Weifang Medical University , Weifang , Shandong , China
Abd-Elhakim Yasmina
Electronic publication date: 2022 Sep 28
Publication date: 2022
Volume: 10
Electronic Location ID: e14086
Received 2022 May 26; Accepted 2022 Aug 29
Copyright: ©2022 Li et al.
Copyright year: 2022
Copyright holder: Li et al.
License: This is an open access article distributed under the terms of the Creative Commons Attribution License, which permits unrestricted use, distribution, reproduction and adaptation in any medium and for any purpose provided that it is properly attributed. For attribution, the original author(s), title, publication source (PeerJ) and either DOI or URL of the article must be cited.
License URL: https://creativecommons.org/licenses/by/4.0/

Keywords: Ac2-26, Hepatic ischemia-reperfusion injury, Apoptosis, Oxidative stress injury, Inflammatory

Funding: Project of Medical and Health Science and Technology 202001020642 The National Natural Science Foundation of China 81000268 Key R&D Program of Shandong Province 2019GSF107056 Key Lab of Shandong Higher Education This work was supported by the development Project of Medical and Health Science and Technology in Shandong Province (202001020642), the National Natural Science Foundation of China (81000268), the Key R&D Program of Shandong Province (2019GSF107056), and the Neurologic Disorders and Regenerative Repair Lab “13th five-year plan” Key Lab of Shandong Higher Education. The funders had no role in study design, data collection and analysis, decision to publish, or preparation of the manuscript.

==============================
Hepatic ischemia-reperfusion injury (HIRI) is one of the major sources of mortality and morbidity associated with hepatic surgery. Ac2-26, a short peptide of Annexin A1 protein, has been proved to have a protective effect against IRI. However, whether it exerts a protective effect on HIRI has not been reported. The HIRI mice model and the oxidative damage model of H2O2-induced AML12 cells were established to investigate whether Ac2-26 could alleviate HIRI by regulating the activation of IL-22/IL-22R1/STAT3 signaling. The protective effect of Ac2-26 was measured by various biochemical parameters related to liver function, apoptosis, inflammatory reaction, mitochondrial function and the expressions of IL-22, IL-22R1, p-STAT3Tyr705. We discovered that Ac2-26 reduced the Suzuki score and cell death rate, and increased the cell viability after HIRI. Moreover, we unraveled that Ac2-26 significantly decreased the number of apoptotic hepatocytes, and the expressions of cleaved-caspase-3 and Bax/Bcl-2 ratio. Furthermore, HIRI increased the contents of malondialdehyde (MDA), NADP+/NADPH ratio and reactive oxygen species (ROS), whereas Ac2-26 decreased them significantly. Additionally, Ac2-26 remarkably alleviated mitochondria dysfunction, which was represented by an increase in the adenosine triphosphate (ATP) content and mitochondrial membrane potential, a decrease in mitochondrial DNA (mtDNA) damage. Finally, we revealed that Ac2-26 pretreatment could significantly inhibit the activation of IL-22/IL22R1/STAT3 signaling. In conclusion, this work demonstrated that Ac2-26 ameliorated HIRI by reducing oxidative stress and inhibiting the mitochondrial apoptosis pathway, which might be closely related to the inhibition of the IL-22/IL22R1/STAT3 signaling pathway.

Introduction

Hepatic ischemia-reperfusion injury (HIRI), a frequently complication due to partial hepatectomy, liver transplantation, hemorrhagic shock, or severe liver trauma surgery, is considered to be one of the most challenging factors determining perioperative morbidity and mortality (Kuboki et al., 2008; Wang et al., 2020). Clinical studies have confirmed that there is a significant correlation between the levels of inflammatory factors and the degree of injury during HIRI (Guo et al., 2016; Jimenez-Castro et al., 2019; Chen et al., 2021). It has been reported that reactive oxygen species (ROS), cytokines and chemokines induced by ischemia-reperfusion injury (IRI) can stimulate immune cells to produce severe inflammation and even cause apoptosis (Granger & Kvietys, 2015; Kan et al., 2018; Jimenez-Castro et al., 2019; Xu et al., 2019; Long et al., 2020). Those data indicated that IRI is actually an inflammatory response dominated by the innate immune response, and blocking the inflammatory response is an ideal strategy to prevent HIRI (Zhang et al., 2014; Konishi & Lentsch, 2017; Yi et al., 2020; Zhang et al., 2020a).

Annexin A1 (AnxA1), an endogenous inflammatory inhibitor, is activated by glucocorticoids, ischemia-reperfusion, inflammation or oxidative stress (Perretti & Flower, 2004; Gerke, Creutz & Moss, 2005). Increasing evidence has shown that the expression of AnxA1 is elevated in IRI of the brain, heart, lung, and retina, and the change of AnxA1 was positively correlated with the degree of IRI (Qin et al., 2019). Ac2-26, a short peptide synthesized from the first 26 amino acids of the N-terminus of AnxA1 protein, retains the anti-inflammatory activity of the AnxA1 (Perretti et al., 2002; Yang et al., 2013). More evidences have substantiated that Ac2-26 not only plays a role in a series of inflammatory diseases, such as endotoxemia, colitis, and arthritis, but also provides protection against IRI of the heart, lung, kidney and brain. However, whether it exerts a protective effect on HIRI has not been reported yet. Based on previous studies and the biological functions of AnxA1 (Yang et al., 1997; Yang et al., 2004; Perretti & D’acquisto, 2009; Guido et al., 2013; Kamaly et al., 2013; Fredman et al., 2015; Li et al., 2019), we speculate that Ac2-26 may attenuate HIRI, but the mechanism is unclear.

Interleukin-22 (IL-22), a member of the IL-10 family, is mainly produced by lymphocytes, such as T helper Th22, Th17, CD8 + T cells, CD4 + T cells, γδ T cells, natural killer (NK) cells, innate lymphoid type 3 (ILC3) cells, and lymphoid tissue inducer cells. IL-22 coordinates immune cells and tissue cells through its functional receptor interleukin-22 receptor 1 (IL-22R1) and plays a key role in tissue damage repair and inflammation regulation in liver, kidney, pancreas, joints and other tissues (Yoon et al., 2006; Yoon et al., 2010). IL-22 acts by binding to heterodimeric receptor complexes, including IL-10R2 and IL-22R1. Since IL-22 has a high affinity for IL-22R1 and a fairly low binding capacity to IL-10R2, we believe that the effect of IL-22 is predominantly/mainly related to the expression of IL-22R1 (Sabat, Ouyang & Wolk, 2014). Primary investigations have showed that IL-22R1 is highly expressed in hepatocytes and epithelial cells, and the expression of IL-22R1 is consistent with the degree of liver injury and inflammation (Chestovich et al., 2012), which suggested that the effect of IL-22 on HIRI was primarily mediated by IL-22R1. It has been reported that IL-22R1, the functional receptor of IL-22, activates signal transducer and activator of transcription 3 (STAT3) by binding to its helical domain. Ki et al. (2010) found that exogenous IL-22 alleviated alcohol-induced liver injury by activating the STAT3 signaling pathway, which was antagonized by neutralizing antibodies against IL-22R1 or STAT3 gene knockout. In addition, IL-22 alleviated ConA-, CCl4-, LPS/D-galactosamine (GAIN)-induced acute liver injury, hepatitis, and fibrosis by activating STAT3. However, Choi et al. (2019) discovered that STX-0119 inhibits the development of liver fibrosis by inhibiting STAT3 and blocking the activation of hepatic stellate cells. Takahashi et al. (2020) found that the IL-22/IL-22R1/STAT3 axis might play a vital role in preventing myocardial post-ischemia reperfusion injury. According to the role of the IL-22/IL-22R1/STAT3 signaling pathway in liver fibrosis, viral hepatitis, IRI of the heart, kidney, and brain (Gavins et al., 2007; Guido et al., 2013; Li et al., 2019; Qin et al., 2019; Huang et al., 2020), we hypothesized that IL-22/IL-22R1/STAT3 signaling might mediate the hepatoprotective effect of Ac2-26.

To verify this hypothesis, we investigated the effects of Ac2-26 on the hepatocyte morphology and function, inflammatory response, oxidative stress, apoptosis, mitochondrial function and the regulation of IL-22/IL-22R1/STAT3 signaling, which might provide a novel scientific basis for the application of Ac2-26 in HIRI therapy.

Materials and Methods

Materials

Ac2-26 was purchased from Qiang Yao Biotechnology Company (Shanghai, China). Hoechst 33342 (HY-15559) and colivelin TFA (CLN, HY-P1061A) were obtained from MedChemExpress (Monmouth Junction, NJ, USA). Hydrogen peroxide (H2O2, 18304), trypan blue (T6146) and rhodamine 123 (Rh123, R8004) were obtained from Sigma (St. Louis, MO, USA). A cell counting kit-8 (CCK-8) colorimetric kit (C008) was purchased from 7sea-biotech (Shanghai, China). Anti-cleaved-caspase-3 (9665) and anti-phospho-STAT3 (p-STAT3Tyr705, 9145) were obtained from Cell Signaling Technology, Inc. (Beverly, MA, USA). Anti-IL-22 (bs-2623R), anti-IL-22R1 (bs-2624R), anti-IL-6 (bs-0782R) and anti-IL-1 β (bs-0812R) antibodies were obtained from Bioss (Beijing, China). Anti-Bax (sc-526) was obtained from Santa Cruz (Santa Cruz, CA, USA). Anti-Bcl-2 (BA0412) was come from Boster (California, USA). Anti-GAPDH (60004-1-Ig) was purchased from Proteintech Group (Chicago, IL, USA). Horseradish-conjugated secondary anti-mouse antibody (ZB-2301) was obtained from ZSGB-BIO (Beijing, China). VECTASHIELD Mounting Medium with DAPI (H−1200 − 10) was purchased from Vector Lab (USA). FITC-conjugated secondary antibody was purchased from Jackson Immunoresearch (805-095-180, West Grove, PA, USA). Malondialdehyde (MDA) assay kit (A003-1-1) was purchased from Nanjing Jiancheng (Nanjing, China). Mouse TNF- α ELISA Kit (EK282/3) was purchased from Multi Science (Hangzhou, China). TdT-mediated dUTP-biotin nick end labeling (TUNEL) assay kit (C1089), NADP +/NADPH detection assay kit (S0179), Bicinchoninic acid (BCA) protein assay kit (P0012) and adenosine triphosphate (ATP) assay kits (S0026) were purchased from Beyotime (Shanghai, China). APC Annexin V apoptosis detection kit with 7-AAD (Annexin V-APC/7-AAD) assay kit (640930) was purchased from Biolegend (San Diego, California, USA). Anti-TNF- α (K009343M), radioimmunoprecipitation assay (RIPA, R0020), phenylmethanesulfonyl fluoride (PMSF, P0100) and ROS assay kit (CA1410) were purchased from Solarbio (Beijing, China). The ReverTra Ace qPCR RT Kit (FSQ-101) was obtained from TOYOBO CO, LTD, Life Science Department (Osaka, Japan). Trizol® (15596026) reagent was purchased from Thermo Fisher Scientific, Inc. (Waltham, MA, USA). DME/F-12 medium (SH30023.01) was obtained from Hyclone (Logan, UT, USA). Nonfat milk (Q/NYLB00395) was purchased from Yili (Neimenggu, China). Super-sensitive enhanced chemiluminescence substrate kit (ECL, ECL-P-100) was obtained from Yanxi Biotechnology Co., Ltd (Shanghai, China).

Animals and treatments

Forty-eight healthy male C57 black 6 (C57BL/6) mice (6–8 weeks old), weighing 22–25 g, SPF grade, were obtained from Pengyue Experimental Animal Center in Jinan. Mice were placed in rearing cages in the experimental animal room and provided with food and water. After housing at constant temperature and humidity with a 12 h light/dark cycle for 7 days, they were randomly divided into sham operation (sham group), ischemia-reperfusion (I/R) group, I/R + Ac2-26 group and Ac2-26 group (n = 8 for each group). Sham and Ac2-26 group mice were subjected to the same anesthesia and surgical procedures, except for occluding the branches of the hepatic pedicle. Ac2-26 (250 µg/kg) or saline was administered intraperitoneally (i.p.) 30 min before ischemia in the I/R + Ac2-26 and I/R groups, respectively. The HIRI model was established as previously described in our laboratory (Wang et al., 2019; Li et al., 2020a). Briefly, mice were anesthetized by intraperitoneal injection of 1% pentobarbital sodium (40 mg/kg), the left and middle branches of the hepatic pedicle were occluded with a vascular clamp for 45 min, followed by 24 h of reperfusion. After reperfusion, mice were deeply anesthetized and sacrificed by cervical dislocation. Blood samples were centrifuged at 4,200 g for 5 min to obtain serum and stored at −80 °C. Part of the left liver tissues were stored at −80 °C, and the other part were fixed in 4% paraformaldehyde. All procedures were performed in accordance with the Animal Ethics Committee of the University (Weifang Medical University Committee for Laboratory Animal Research, which provided full approval for this research, No. 2020SDL187).

Cell culture and treatment

Mouse hepatocyte AML12 cells were purchased from the Chinese Academy of Sciences Cell Bank (RRID: CVCL_0140, Shanghai, China). AML12 cells were incubated at 37 °C in a humidified atmosphere containing 5% CO2. The cells were divided into six groups: control group, H2O2 group, H2O2 + Ac2-26 group, Ac2-26 group and H2O2 + Ac2-26 + CLN group. Cell death was induced by 200 µM H2O2 according to the previous literature from our laboratory (Wang et al., 2019; Li et al., 2020a). In brief, cells were challenged with H2O2 with or without Ac2-26 for 6 h. Cells in the H2O2 + Ac2-26 + CLN group were pretreated with CLN (1 µM, Fig. S1) 1 h before Ac2-26 treatment.

Trypan blue dye exclusion assay

AML12 cells treated with or without Ac2-26 were stained in 0.4% trypan blue in phosphate buffered saline (PBS) for 3 min, washed two times with PBS, and then observed under a phase-contrast microscope (Leica Microsystems, Germany) (Sun et al., 2022).

Cell viability assay

A CCK-8 colorimetric kit was used to detect cell viability (Li et al., 2020a). AML12 cells were adjusted to 1 × 105 cells/mL, seeded in 96-well plates and cultured overnight at 37 °C. Subsequently, cells were cultured in a solution of 100 µL of CCK-8/mL DME/F-12 for 1 h in a CO2 incubator in the dark. The optical density (OD) at 450 nm was measured by a microplate reader (Multiskan FC; Thermo Fisher, Waltham, MA, USA). The experiment had 3 replicate wells in each group and was repeated five times. The cell viability was calculated according to the following formula: Cell viability=the absorbance of the experimental group/the absorbance of the control group.

Histopathological examination

Perfused livers were fixed with 4% paraformaldehyde for 48 h and embedded in paraffin, and serial 5- µm- thick sections were cut. Tissue sections were stained with hematoxylin and eosin in a standard manner dehydrated in graded alcohols and xylene and mounted with neutral gum. The morphological changes in hepatocytes were observed under light microscopy, and the Suzuki score (Suzuki et al., 1993; Cienfuegos-Pecina et al., 2021) was used to evaluate the degree of liver injury: 0 = no congestion, no vacuolization, no necrosis; 1 = minimal congestion, minimal vacuolization, single-cell necrosis; 2 = mild congestion, mild vacuolization, necrosis <30%; 3 = moderate congestion, moderate vacuolization, necrosis <60%; 4 = severe congestion, severe vacuolization, necrosis >60%.

TUNEL staining

After deparaffinization and hydration, sections were digested with 20 µg/mL Proteinase K for 20 min at room temperature, and blocked with hydrogen peroxide for 10 min to inactivate endogenous peroxidases. Then sections were incubated with TUNEL reaction mixture in a dark room for 2 h at 37 °C, blocked with blocking solution for 30 min at room temperature, covered with mounting medium with DAPI and observed under a fluorescence microscope (Olympus FV500, Japan) (Zhao et al., 2019).

Quantitative real-time PCR (qRT–PCR)

Total RNA of the liver tissues (100 mg) was isolated using TRIzol® reagent, and cDNA was reversed transcribed from total RNA using a ReverTra Ace qPCR RT Kit according to the manufacturer’s protocol. PCR amplification was performed as previously reported (Fan et al., 2007). In brief, the cDNA was pre-denatured at 95 °C for 10 min, followed by 45 cycles of 95 °C for 30 s, 60 °C for 30 s, and extended at 72 °C for 30 s. The primer sequences for IL-22, IL-22R1, mtApt6, Rp113, COX-1, ND1 and GAPDH are summarized in Table 1. GAPDH was used as the internal control. The relative level of target genes was normalized against GAPDH. The mtDNA copy number was calculated by the 2−ΔΔCt method.

Table 1 Nucleotide sequences of primers used for qRT-PCR.

Gene	Sequence	
IL-22	Forward: 5′-TCCAACTTCCAGCAGCCATACATC-3′	
Reverse: 5′-GCACTGATCCTTAGCACTGACTCC-3′	
IL-22R1	Forward: 5′- CGTCAACCACACCTACCAGATGC -3′	
Reverse: 5′- AGC GTC GAG CCG AGG AACTC -3′	
mtAtp6	Forward: 5′- GCCGTAATTCTAGGCTTCCGACAC -3′	
Reverse: 5′- TGCTGTTAGTCGTACTGCTAGTGC -3′	
Rp113	Forward: 5′- GCAGCCTTTGCATCATTGTCCTTC- 3′	
Reverse: 5′- CCCAGGCGGCTTTAGACAAGTAAC- 3′	
COX-1	Forward: 5′- TCAACCTTGTCAACACAGCCTCAC - 3′	
Reverse: 5′- GGCACACGGAAGGAAACATAGGG - 3′	
ND1	Forward: 5′- GCACCAGCCCTTCCTTTGACG - 3′	
Reverse: 5′- TCGGCGGATGGTTCGTGTTTG - 3′	
GAPDH	Forward: 5′-TGATTCTACCCACGGCAAGTT-3′	
Reverse: 5′-TGATGGGTTTCCCATTGATGA-3′	

Western blotting

Tissues or cells were homogenized on ice in RIPA lysis buffer with protease inhibitor mix PMSF. Lysates were obtained by centrifugation at 16,800 g for 15 min at 4 °C. The protein concentration was determined by the BCA method. Next, equal protein amounts were subjected to electrophoresis in 10% polyacrylamide gels and then transferred to PVDF membranes by electro-blotting. The membranes were blocked with 5% nonfat milk and subsequently incubated overnight at 4 °C with primary antibodies such as anti-IL-22 (1:500), anti-IL-22R1 (1:500), anti-cleaved-caspase-3 (1:1000), anti-Bax (1:500), anti-Bcl-2 (1:400), anti-p-STAT3Tyr705 (1:1000) and anti-GAPDH (1:1000), followed by labeling with horseradish peroxidase-conjugated secondary antibodies and detection with an ECL detection system (ChemiDoc™ Touch, Bio Rad, USA). Finally, the images were analyzed by ImageJ (National institutes of Health, USA) software. GAPDH was used as the internal reference protein. The results were normalized to the ratio of the target protein to the internal reference protein GAPDH. Each experiment was repeated at least three times (Zhang et al., 2020b).

Enzyme-linked immunosorbent assay (ELISA) analysis

The levels of tumor necrosis factor- α (TNF- α) in the serum were measured using commercial kits according to the manufacturer’s instructions. Plates were read using a microplate reader at 450 nm excitation and 510 nm emission (Multiskan FC, Thermo, Massachusetts, USA) (Zhao et al., 2019).

MDA assays

MDA of liver tissue was measured with commercial kits according to the manufacturer’s recommendations as previously described (Yu et al., 2013; Chen et al., 2020). MDA was expressed as nM/mg protein. The level of MDA was calculated using the following formula: MDA content in tissue (nM/mg prot)=[(the OD of experimental group –the OD of control group)/(the OD of standard group –the OD of blank group)] × standard product concentration (10 nM/mL)/sample protein concentration (mg prot/mL).

NADP+/NADPH ratio determined

The NADP+/NADPH ratio was determined using the NADP+/NADPH Assay Kit. According to the manufacturer’s protocols and the method described previously (Xie et al., 2020; Wen et al., 2021). Briefly, about 20 mg of hepatic tissue was dissected and cut into small pieces, added to 400 µL of NADP+/NADPH extraction buffer, homogenized on ice, and then centrifuged at 12,000 g for 10 min at 4 °C, and the supernatant was taken as the sample to be tested. To detect the NADPtotal, 50 µL of sample and 100 µl of G6PDH working solution were added sequentially to a 96-well plate. After incubation at 37 °C for 10 min in the dark, 10 µL of chromogenic solution was added, following another 20 min incubation at 37 °C, the OD value was determined by a microplate reader (Multiskan FC, 168 Thermo, Massachusetts, USA) at 450 nm. To assess the content of NADPH, the samples (200 µL) were incubated in the water bath at 60 °C for 30 min. The OD value at 450 nm was detected according to the above steps. The NADP+/NADPH ratio was calculated according to the standard curve and following formula: [NADP+]/[NADPH]=([NADP total]−[NADPH])/[NADPH].

Enhanced ATP assay

A commercially available ATP assay kit was used to determine the intracellular ATP level (Zhang et al., 2019a). Lysed AML12 cells were collected in a 1.5 mL EP tube and centrifuged at 16,800 g for 5 min at 4 °C. Subsequently, the supernatant was mixed with the luciferase reagent in an opaque 96-well plate, and the OD value was measured with a luminometer.

Annexin V-APC/7-AAD assay

After digested by pancreatin and washed twice with cold PBS, cells were suspended in 100 µL Annexin V binding buffer, and incubated with 5 µL Annexin V-APC and 5 µL 7-AAD for 15 min at room temperature in the dark. 400 µL Annexin V binding buffer was added to each sample tube (Sai et al., 2021). The samples were detected by FACS (BD FACSAria3, Becton-Dickinson, San Jose, CA, USA).

Hoechst 33342 staining

Hoechst 33342 can be used to observe nuclear morphologic. In healthy cells, the nucleus is spherical, and DNA is evenly distributed. During apoptosis, the DNA becomes condensed and fragmented. AML12 cells were cultured on glass slides overnight at 37 °C, pre-treated cells were washed twice with PBS, stained with Hoechst 33342 for 15 min, washed twice with PBS, covered with mounting medium and observed the morphologic of nuclei under a 350 nm filter with a fluorescence microscope (Ye et al., 2017) (Olympus FV500; Olympus, Tokyo, Japan).

Rhodamine 123 staining

Mitochondrial membrane potential was detected by Rhodamine 123 (Rh123), a kind fluorescent dye that can selectively stain mitochondria of living cells (Wei et al., 2020). AML12 cells of various groups were incubated with 2 µM Rh123 for 40 min at 37 °C in the dark, rinsed with PBS, covered with mounting medium with DAPI and observed under a fluorescence microscope (Olympus FV500, Japan).

To evaluate the level of membrane potential in H2O2-induced AML12 cells with or without Ac2-26, the fluorescence intensity of Rh123 was also investigated using a FACS Calibur flow cytometer (BD FACSAria3; Becton-Dickinson, San Jose, CA, USA) with an excitation wavelength of 485 nm and emission wavelength of 530 nm. The results are expressed as the fold change of the percentage increase in the Rh123 channel. The experiments were performed in triplicate and repeated three times.

Detection of ROS level

ROS were assessed by 2′,7′-dichlorodihydrofluorescein diacetate min in the dark at 37 °C in PBS containing 10 µM H2DCFDA and then (H2DCFDA) staining as previously described (Jiang et al., 2014; Li et al., 2020a). Briefly, AML12 cells from different groups were collected and incubated for 20 detected by flow cytometry (BD FACSAria3, Becton-Dickinson, San Jose, CA, USA) to determine the changes in ROS content using a method similar to Rh123 staining.

Statistical analysis

We used SPSS 25.0 software (IBM Corp. Armonk, NY, USA) for data analysis and GraphPad Prism 7 software (San Diego, CA, USA) for drawing. All data are presented as means ± standard error of the mean (SEM) of at least triplicate samples. Variance homogeneity test were performed. One-way analysis of variance (ANOVA) was used to compare multiple groups, followed by the least significant difference (LSD) test. A P-value of 0.05 was considered statistically significant.

Result

Ac2-26 improved the viability of AML12 cells under H2O2 stress

Similar to AnxA1, Ac2-26 could also regulate endogenous inflammation. To clarify the effect of Ac2-26 on HIRI, we observed the effect of Ac2-26 on the oxidative damage model of AML12 cells induced by H2O2. According to our previous study, 200 µM H2O2 was used to induce oxidative stress injury in AML12 cells (Wang et al., 2019; Li et al., 2020a). Preliminary test results showed that 0.8 µM Ac2-26 has the strongest protective effect on AML12 cells (Fig. S2). Therefore, 0.8 µM Ac2-26 was used for subsequent experiments.

To determine whether Ac2-26 could reduce H2O2-induced oxidative stress in AML12 cells, cell viability was investigated by CCK-8 assay. As shown in Fig. 1A, compared with the control group, the cell viability (−63.72%) was significantly reduced in the H2O2 group (∗P = 0.0155), which was reversed by Ac2-26 pretreatment (2.57-fold higher, ∗P = 0.0277). Similar to the CCK-8 results, trypan blue staining showed that Ac2-26 alleviated the H2O2-induced AML12 cell death (−70.90%, ∗∗∗P < 0.001) (Figs. 1B, 1C). The above results suggested that Ac2-26 could improve the viability of AML12 cells under H2O2 stress.

Figure 1 Ac2-26 pretreatment improved the viability of AML12 cells under H2O2 stress.

(A) CCK-8 assay for cell viability (n = 3). (B–C) The death rate was assessed by trypan blue staining (×200, n = 5). Note that Ac2-26 pretreatment attenuated the H2O2-induced changes in cell viability and cell death rate. The arrows indicate dead cells. Data are represented as means ± SEM, ∗P < 0.05, ∗∗∗P < 0.001, ∗∗∗∗P < 0.0001, ns P > 0.05.

Ac2-26 protected the morphology and function of liver cells after HIRI

According to the results of the preliminary experiment, 250 µg/kg Ac2-26 significantly protected the HIRI model in mice (Fig. S3). Next, we observed the effect of Ac2-26 on hepatocytes after ischemia-reperfusion. Hematoxylin-eosin (HE) staining showed that the structure of hepatic lobules in the I/R group was severely damaged, with vacuolization, congestion and sheet necrosis accompanied by inflammatory cell infiltration. While, these pathological changes were significantly decreased in the I/R + Ac2-26 group. As shown in Fig. 2A, the Suzuki score showed a sensible increase (by 6.36-fold) in histological score in the I/R group compared with the sham group (∗∗∗∗P < 0.0001), which was reversed by Ac2-26 pretreatment (−64.15%, ∗∗∗P < 0.001) (Fig. 2B). The results indicated that Ac2-26 could ease the structural destruction of hepatocytes induced by ischemia-reperfusion.

Figure 2 Ac2-26 improved HIRI -induced liver injury.

(A) Histopathological examination by HE staining (×200). The dashed area represents the necrotic area. And the vacuolization and inflammatory cell infiltration are indicated by arrows and triangles, respectively. (B) Suzuki score assessed the degree of liver tissue damage (n = 3). The results showed that Ac2-26 could improve the structure of hepatocytes and decrease the Suzuki score. Data are represented as means ± SEM, ∗∗∗P < 0.001, ∗∗∗∗P < 0.0001, ns P > 0.05.

Ac2-26 attenuated the inflammatory reaction induced by HIRI

To determine the effect of Ac2-26 on inflammatory response induced by HIRI, we investigated the expression of TNF- α, IL-6 and IL-1 β. Western blotting analysis showed that the expressions of TNF- α (by 1.44-fold, ∗∗P = 0.0018), IL-6 (by 1.63-fold, ∗∗P <0.0019) and IL-1 β (by 1.48-fold, ∗∗P = 0.0062) were significantly increased in the I/R groups compared with the sham groups, which was reversed by preconditioning of Ac2-26 (TNF- α: −21.09%, ∗P = 0.0218; IL-6: −23.50%, ∗P = 0.0207; IL-1 β: −26.10%, ∗P = 0.0316) (Figs. 3B–3D). To further clarify the effect of Ac2-26 on the release of inflammatory factors, the serum TNF- α level was measured with ELISA assay. Compared with the sham group, the TNF- α content (by 3.24-foid) was significantly increased in the I/R group ( ∗∗P = 0.0015), while significantly decreased in the I/R + Ac2-26 group (−45.82%, ∗P = 0.0282) (Fig. 3A).

Figure 3 Ac2-26 attenuated the elevated levels of inflammatory cytokine induced by HIRI.

(A) The serum TNF- α level was measured with ELISA (n = 7). (B–D) The expressions of TNF- α (n = 6), IL-6 (n = 5) and IL-1 β (n = 8) proteins in liver tissues. Results showed that the expression and secretion of inflammatory cytokine had a significant up-regulation in HIRI, and Ac2-26 pretreatment rescued those changes. Data are represented as means ± SEM, ∗P < 0.05, ∗∗P < 0.01, ns P > 0.05.

Ac2-26 reduced HIRI-induced apoptosis in vivo and in vitro

Studies have shown that apoptosis is involved in ischemia-reperfusion injury. TUNEL staining, Hoechst 33342 staining, Annexin V-APC/7-AAD assay and western blotting were used to evaluate the effect of Ac2-26 on apoptosis induced by HIRI. The results showed that apoptotic cells were increased in the I/R group and H2O2 group (by 2.28-foid, ∗∗P = 0.0047, Figs. 4A–4C), along with a higher level of cleaved-caspase-3 (by 1.48-fold, ∗∗P = 0.0078) expression and Bax/Bcl-2 ratio (by 2.01-fold, ∗∗∗P < 0.001) (Figs. 4D, 4E), while Ac2-26 antagonized these changes (C: −37.57%, ∗P = 0.0361; D: −31.07%, ∗∗P = 0.0083; E: −39.18%, ∗∗P = 0.0018). All data proposed that Ac2-26 could inhibit HIRI-induced apoptosis.

Figure 4 Ac2-26 reduced HIRI-induced apoptosis.

(A) Apoptotic nuclei were detected by TUNEL staining (red, ×200, in vivo). (B) Nuclei were detected by Hoechst 33342 staining (blue, ×200, in vitro). (C) Annexin V-APC/7-AAD assay analysis of cell apoptosis (n = 5, in vitro. Q1 portion is early apoptotic cells, Q2 portion is late apoptotic cells.). (D) Western blotting analysis of the apoptotic protein cleaved-caspase-3 (n = 8, in vivo). (E) Western blotting analysis of apoptotic proteins: Bax (n = 5, in vivo) and Bcl-2 (n = 5, in vivo). Data are represented as means ± SEM, ∗P < 0.05, ∗∗P < 0.01, ∗∗∗P < 0.001, ns p > 0.05.

Ac2-26 attenuated the level of oxidative stress in vivo and in vitro

Oxidative stress injury is an important mechanism of HIRI. Subsequently, the intracellular ROS, MDA and NADP +/NADPH ratio were detected with commercial kits. Our results showed that the MDA content, an end product of lipid peroxidation, in the liver homogenates of the I/R group was 5.92 ± 1.25 µM/g prot, which was increased 3.13-fold than that in the sham group (1.89 ± 0.24 µM/g prot), whereas Ac2-26 pretreatment significantly attenuated this increase (−57.19%, ∗∗P = 0.0023) (Fig. 5A). The ratio of NADP +/NADPH was significantly increased in the I/R groups compared with the sham groups (by 14.84-fold, ∗∗∗∗P < 0.0001), which was decreased by preconditioning of Ac2-26 (−76.49%, ∗∗∗∗P < 0.0001). Furthermore, in vitro, we detected the levels of intracellular ROS in AML12 cells by flow cytometry assay. In agreement with the results of MDA, incubation of AML12 cells with H2O2caused a significant increase in ROS compared with untreated cells (by 5.82-fold, ∗P = 0.0314) (Fig. 5C). The observation suggests that Ac2-26 can decrease oxidative stress reactions in vivo and in vitro.

Figure 5 Ac2-26 attenuated the level of oxidative stress in vivo and in vitro.

(A) Ac2-26 reversed the increase in MDA content induced by HIRI ( n = 5). (B) Ac2-26 reversed the increase in NADP +/NADPH ratio induced by HIRI (n = 4). (C) The ROS of AML12 cells was detected by flow cytometry (n = 5, Rosup represents positive control, and all groups are stained with H2DCFDA). The results showed that the increased in MDA, NADP +/NADPH ratio, and ROS content were significantly reduced by incubation with Ac2-26. Data are represented as means ± SEM, ∗P < 0.05, ∗∗P < 0.01, ∗∗∗P < 0.001, ∗∗∗∗P < 0.0001, ns P > 0.05. as means ± SEM, ∗P < 0.05, ∗∗P < 0.01, ∗∗∗∗P < 0.0001, ns P > 0.05.

Ac2-26 ameliorated mitochondrial damage induced by H2O2

It is reported that the mtApt6/Rp113 ratio and the COX-I and ND1 expression may be used to represent the content and transcription level mtDNA (Wang et al., 2019; Li et al., 2020). The results of qRT-PCR showed that the ratio of mtAtp6/Rp113 was significantly increased (by 1.66-fold, ∗P = 0.0192), and the expressions of COX-1 (−46.35%, ∗P = 0.0291) and ND1 (−43.78%, ∗P = 0.0227) were decreased in the H2O2 group compared with the control group. Ac2-26 pretreatment antagonized these changes (A: −36.82%, ∗P = 0.0265; B: by 1.90-fold, ∗P = 0.0249; C: by 1.66-fold, ∗P = 0.0452) (Figs. 6A–6C). The above changes suggested that Ac2-26 could maintain the function of mitochondria by stabilizing the quality and quantity of mtDNA.

Figure 6 Ac2-26 ameliorated mitochondrial damage induced by H2O2.

(A–C) The mRNA levels of mtApt6/Rp113, COX-1 and ND1 (n = 3). (D) Flow cytometry detected mitochondrial membrane potential (n = 4, 75% ethanol represents negative control). (E, F) Mitochondrial membrane potential observed by Rh123 staining under fluorescence microscopy (n = 5). (G) The content of ATP in AML12 cells (n = 6). Note that Ac2-26 pretreatment attenuated the H2O2-induced mitochondrial dysfunction represented by reduced mtDNA damage and increased ATP content and mitochondrial membrane potential. The arrows represent the rhodamine 123 fluorescence became diffuse because of mitochondrial depolarization. Data are represented as means ±  SEM, ∗P < 0.05, ∗∗P < 0.01, ∗∗∗∗P < 0.0001, ns P > 0.05.

As a sensitive indicator of mitochondrial function, collapse of mitochondrial membrane potential is related to mitochondrial dysfunction, or even cell death (Hengartner, 2000). We therefore investigated mitochondrial membrane potential using Rh123 staining. As shown in Figs. 6D–6F, the fluorescence intensity of Rh123 in the H2O2 group showed a significant decrease compared to the untreated group (D: −27.98%, ∗∗P = 0.0046; F: −68.20%, ∗∗∗∗P < 0.0001). While the fluorescence intensity in Ac2-26-treated group was significantly increased compared to H2O2 group (D: by 1.27-fold, ∗P = 0.0463; F: by 2.35-fold, ∗∗∗∗P < 0.0001).

To further explore the effect of Ac2-26 on mitochondrial function induced by HIRI, the ATP content was measured using an H2O2-induced oxidative damage model with or without Ac2-26. The ATP content was markedly lower in the H2O2 group than that in the control group (−37.20%, ∗∗P = 0.0084), while it was reversed by Ac2-26 pretreatment (by 1.25-fold, ∗P = 0.0233) (Fig. 6G), suggesting that Ac2-26 could rescue mitochondrial function.

Taken together, these data indicated that Ac2-26 could alleviate H2O2-induced mitochondrial dysfunction, which was manifested by increasing ATP content, inhibiting the loss of mitochondrial membrane potential and regulating the balance of mtDNA.

Ac2-26 inhibited the expressions of IL-22 and its receptor IL-22R1 induced during HIRI

As an important regulating factor of inflammation, IL-22 can regulate the inflammatory response and repair tissue damage through IL-22/IL-22R1. To investigate the mechanism of the cytoprotective effect of Ac2-26 on hepatocytes, the expressions of IL-22 and IL-22R1 were determined by qRT–PCR and western blotting. The results of qRT–PCR indicated that the levels of IL-22 (by 9.08-fold) and IL-22R1 (by 1.93-fold) in the I/R group were significantly increased compared to those in the sham group (∗∗∗P < 0.001; ∗∗P = 0.0021), and Ac2-26 effectively blocked these changes induced by HIRI (IL-22: −43.41%, ∗P = 0.0276; IL-22R1: −48.13%, ∗∗P = 0.0043) (Figs. 7A, 7B). These facts were further confirmed by western blotting (C, ∗∗P = 0.0052, ∗∗P = 0.0011; D, ∗∗P = 0.0019, ∗∗P = 0.0090) (Figs. 7C, 7D).

Figure 7 Ac2-26 inhibited the expressions of IL-22 and IL-22R1 induced by HIRI.

(A–B) The mRNA levels of IL-22 and IL-22R1 (n = 5). (C–D) The expressions of IL-22 (n = 4) and IL-22R1 ( n = 5) proteins in liver tissues. Note the significantly increased IL-22 and IL-22R1 expressions induced by HIRI were rescued by Ac2-26 administration. Data are represented as means ± SEM, ∗P < 0.05, ∗∗P < 0.01, ∗∗∗P < 0.001, ns P > 0.05.

Ac2-26 prevented the activation of STAT3 induced by HIRI

A large amount of evidence indicates that IL-22 alleviates IRI in the heart, brain, liver, kidney and other organs through STAT3 signaling. To determine whether the hepatoprotective effect of Ac2-26 is related to the inhibition of the IL-22/IL-22R1/STAT3 signaling pathway, the expression of p-STAT3Tyr705 was measured by western blotting. As shown in Fig. 8A (by 7.28-fold) and 8B (by 2.65-fold), the expression of p-STAT3Tyr705 was significantly increased in HIRI group compared with the sham group (∗∗∗P < 0.001; ∗∗∗P < 0.001), which was significantly inhibited by administration of Ac2-26 (A: −40.45%, ∗P = 0.0268; B: −37.65%, ∗P = 0.0259). In summary, our present study demonstrated that STAT3 signaling was activated during HIRI and Ac2-26 could inactivate this signal, which suggested that Ac2-26 could inhibit HIRI-induced activation of the STAT3 pathway.

Figure 8 Ac2-26 prevented the activation of STAT3 induced by HIRI.

(A–B) The expression of p-STAT3Tyr705 in vivo (n = 6) and in vitro (n = 6). The expression of p-STAT3Tyr705 was up-regulated in HIRI and Ac2-26 administration inhibited it significantly. Data are represented as means ± SEM, (n ≥ 3), ∗P < 0.05, ∗∗∗P < 0.001, ns P > 0.05.

To further verify whether the protective effect of Ac2-26 on hepatocytes was related to the STAT3 signaling pathway, we next observed the effect of CLN, a STAT3-specific activator, on the hepatoprotection of Ac2-26. As shown in Fig. 9A, Ac2-26 significantly attenuated the morphological change in hepatocytes, and CLN intervention abolished the protective effect of Ac2-26. Similarly, the CCK-8 assay (Fig. 9B, ∗∗∗P < 0.001, ∗P = 0.0181, ∗P = 0.0137 respectively) and trypan blue staining (Figs. 9C, 9D, ∗∗∗∗P <0.0001, ∗∗P = 0.0010, ∗P = 0.0370 respectively)) showed that CLN antagonized the effect of Ac2-26 on cell activity (−30.40%) and the death rate (by 2.48-fold) induced by H2O2. These data further confirmed that the IL-22/IL-22R1/STAT3 signaling pathway was involved in hepatocyte damage during HIRI.

Figure 9 CLN antagonized the protective effect of Ac2-26 on oxidative stress damage in AML12 cells.

(A) Inverted microscopy was used to observecell morphology ( ×200). (B) CCK-8 assay for cell viability (n = 4). (C–D) Trypan blue staining was used to assess the cell death rate (×200, n = 5). Note that the protective effect of Ac2-26 was reversed by CLN, a STAT3-specific activator. The arrows indicate dead cells. Data are represented as means ± SEM, ∗P < 0.05, ∗∗P < 0.01, ∗∗∗P < 0.001, ∗∗∗∗P < 0.0001, ns P > 0.05.

Discussion

In this study, the hepatoprotective effect of AC2-26 and its possible mechanism were investigated in vivo and in vitro, and two major findings were obtained. First, ischemia-reperfusion produced remarkable injuries in hepatocytes, such as the morphological damage, dysfunction, increased molecular levels of inflammatory response, elevated level of oxidative stress, hepatocyte apoptosis with mitochondrial dysfunction, and the activation of the IL-22/IL-22R1/STAT3 signaling pathway, which suggested that the IL-22/IL-22R1/STAT3 signaling mediated the mitochondrial apoptosis pathway during HIRI. Second, we also discovered that the administration of Ac2-26 significantly inhibited apoptosis and the activation of IL-22/IL-22R1/STAT3 signaling, whereas CLN antagonized the effect of Ac2-26, which suggested that the hepatoprotective effect of Ac2-26 was related to the IL-22/IL-22R1/STAT3 signaling pathway.

Previous studies in our team indicated that ischemia-reperfusion not only induced significant hepatocyte injury but also led to the increased expression of inflammatory factors such as NF- κB, TNF- α, TLR4, IL-1 β and NLRP3, as well as concomitant hepatocyte apoptosis, ATP depletion, and the loss of mitochondrial membrane potential, etc. The data demonstrated that the inflammatory response and mitochondrial apoptosis pathway were involved in the process of HIRI (Yu et al., 2013; Jiang et al., 2014; Wang et al., 2019; Li et al., 2020a).

As an endogenous anti-inflammatory protein regulated by glucocorticoids, AnxA1 is expressed at low levels in the cytosol, elevated in response to ischemia-reperfusion, oxidative stress and glucocorticoids, and then translocated from the cytosol to the membrane, inhibiting leukocyte migration to inflammation sites and reducing the inflammatory response. Therefore, AnxA1 mainly exerts anti-inflammatory effects by interfering with the aggregation, migration, and adhesion of neutrophils (Perretti & D’acquisto, 2009). Ac2-26, an active N-terminal peptide of AnxA1, has been widely used in various diseases, such as acute inflammation, chronic inflammation and IRI etc., as an alternative to AnxA1 (Damazo et al., 2006; Gavins et al., 2007). Mounting evidence has shown that the administration of Ac2-26 in lung IRI significantly attenuates pulmonary edema, pro-inflammatory cytokine production, oxidative stress, apoptosis, neutrophil infiltration, and pulmonary tissue injury, which indicates that Ac2-26 exerts a protective effect against acute pulmonary injury induced by I/R (Liao et al., 2017; Gong et al., 2019; Machado et al., 2020). Other studies have shown that Ac2-26 decreases pro-inflammatory mediators and increases anti-inflammatory cytokines during pulmonary injury induced by intestinal I/R and obstructive pulmonary disease models (Guido et al., 2013; Possebon et al., 2018). It has also been reported that Ac2-26 treatment exerts protective effects on IRI of the heart, brain, kidney, retina and other organs by inhibiting neutrophil infiltration and the relationship of pro-inflammatory cytokines (Facio et al., 2011; Girol et al., 2013; Yang, Morand & Leech, 2013; Qin et al., 2015; Yu et al., 2019; Zhao et al., 2019; Xu et al., 2021). However, whether Ac2-26 has a protective effect on HIRI has not been reported yet. Our results highlighted that Ac2-26 attenuated HIRI by inhibiting inflammation, oxidative stress and mitochondrial apoptosis pathway, as characterized by a decrease in MDA content, TNF- α, IL-6 and IL-1 β levels, number of apoptotic cells, expression of cleaved-caspase-3 and imbalance of anti- and pro-apoptotic proteins in Bcl-2 family proteins.

Although little is known about the mechanism of HIRI, it is becoming clear that the local inflammatory response induced by ischemia-reperfusion is currently considered to be the main cause of cell death. As a member of the IL-10 superfamily, IL-22 plays a role in regulating inflammation by forming compound of IL-22/IL-22R1/IL-10R2 compounds, which subsequently activate STAT3 signaling (Xie et al., 2000). In the liver, hepatocytes are the main target cells for IL-22 action. Numerous studies have indicated that IL-22 administration activates hepatic STAT3 signaling and ameliorates several liver diseases, including autoimmune hepatitis, viral hepatitis, drug-induced liver injury, alcoholic fatty liver and steatohepatitis (Dumoutier et al., 2009; Ki et al., 2010; Wolk et al., 2010). Chestovich and Radaeva et al. (Radaeva et al. 2004; Chestovich et al., 2012) confirmed that the expression of IL-22R1 was increased during hepatic ischemia-reperfusion and the recombinant IL-22 alleviated morphological destruction and functional damage of hepatocytes, along with the expressions of IL-22R1 and pro-inflammatory cytokines, which suggested that IL-22 protein exerted hepatoprotection by STAT3 activation. Based on previous studies, we explored the role of IL-22/IL-22R1/STAT3 signaling during HIRI. We found that the expression levels of IL-22, IL-22R1 and p-STAT3Tyr705 were significantly increased during HIRI, which revealed ischemia-reperfusion activated IL-22/IL-22R1/STAT3 signaling. On this basis, we speculated that the activation of the IL-22/IL-22R1/STAT3 signaling pathway caused by HIRI may be an auto-regulatory response of the organism, but continued excessive activation may cause further damage to the organism.

All of these findings suggested that IL-22/IL-22R1/STAT3 signals may be involved in the development of HIRI, however, the role of activated STAT3 in HIRI remains controversial (Sima & Ma, 2019; Takahashi et al., 2020; Zhao et al., 2020). Han et al. (2018) discovered that resveratrol provided neuroprotection by activating the JAK2/STAT3 pathway. Sima & Ma (2019) found that sevoflurane could significantly improve HIRI and reduce hepatic immune inflammation by activating the JAK2/STAT3 pathway. Yang et al. (2013) proved that melatonin pretreatment could decrease the mitochondrial oxidative damage induced by myocardial ischemia-reperfusion by intervention with H03867, an inhibitor of STAT3, or STAT3-siRNA could aggravate the activating the JAK2/STAT3 signaling pathway. Han et al. (2018) revealed that the degree of hepatocyte apoptosis and HIRI, whereas the over-expression of STAT3 could ameliorate HIRI. Radaeva et al. (Radaeva et al., 2004; Dong et al., 2021) showed that recombinant IL-22 or overexpression of IL-22 in hepatocytes could activate STAT3 and induce the expressions of a variety of anti-apoptotic factors (e.g., Bcl-2, Bcl-xL, Mcl-1), whereas IL-22 or STAT3 blocked could abolish the anti-apoptotic action of IL-22 in hepatocytes, which demonstrated that IL-22 could protect against hepatic damage by activating the JAK2/STAT3 signaling pathway (Radaeva et al., 2004; Dong et al., 2021), which revealed that up-regulation or activation of STAT3 was one of the pivotal molecular mechanisms of HIRI. In contrast, Sun et al. (2020) has found that AG490 or JAK2 siRNA significantly alleviated cerebral ischemic injury. Li and Chen (Chen et al., 2020; Li et al., 2020b) demonstrated that stachydrine and matrine could improve nerve function and ameliorate brain edema and apoptosis by inhibiting the aberrant activation of the JAK2/STAT3 signaling pathway. Similarly, researches also showed that drug preconditioning inhibited IRI in the kidney, heart and other organs by down-regulating aberrant activation of the JAK2/STAT3 signaling pathway (Luo et al., 2016; Zhang et al., 2020b; Zhao et al., 2020). Regarding the role of STAT3 signaling in liver disease, Xiong et al. (2018) observed that the inhibition of STAT3 attenuated HIRI. Zhang et al. (2019b) discovered that magnesium lithospermate B protected against HIRI by inhibiting the JAK2/STAT3 signaling pathway and decreasing the aberrant secretions of IL-6, IL-1 β, TNF- α and other inflammatory cytokines. The aforementioned studies demonstrate that the excessive activation of JAK2/STAT3 signaling is involved in the development of IRI, and various drug interventions can regulate the activation of JAK2/STAT3 signaling to protect against IRI. Our findings unraveled that Ac2-26 pretreatment inhibited the HIRI-induced expressions of IL-22, IL-22R1 and p-STAT3Tyr705, whereas the intervention with CLN antagonized the effects of Ac2-26, which revealed that the hepatoprotective effect of Ac2-26 was related to the IL-22/IL-22R1/STAT3 signaling pathway.

Figure 10 The possible mechanism of Ac2-26 hepatoprotection during HIRI.

The ischemia-reperfusion injury induced the secretion of IL-22 by Th cell and NK cell. The action of IL-22 was mediated by binding to the heterodimeric receptors of IL-22R1 and IL-10R2, which could phosphorylate the JAK2/STAT3 signaling. Subsequently, p-STAT3 molecule appeared to be dimerized, translocated into the nucleus, and bound to specific DNA sequence to induce the transcription of many gens that regulating apoptosis and proliferation. Ac2-26 could reduce ischemia-reperfusion induced hepatocytes damage via inhibiting the secretion of IL-22 and inhibition of the JAK2/STAT3 signaling pathway.

Conclusions

In conclusion, the data presented in this study provide preliminary evidence demonstrating for the protective effects of Ac2-26 against HIRI. As shown in Fig. 10, the protection of Ac2-26 may be conferred by one or more of the following mechanisms. First, Ac2-26 could reduce the expressions of IL-22 and its receptor IL-22R1. Second, Ac2-26 could inhibit the mitochondrial apoptosis pathway, as measured by cell apoptosis, mitochondrial dysfunction and oxidative stress, etc. Third, the beneficial effects of Ac2-26 might be associated with inhibition the over activation of STAT3. However, the current study still had some limitations; due to the limitations of the laboratory environment and conditions, the present study failed to study the protective effect of Ac2-26 using STAT3 KO mice or mouse primary hepatocyte. Nevertheless, the present study provided evidence and laid the foundation for clinical applications of Ac2-26 and might help in developing more effective therapy for HIRI.

Supplemental Information

Supplemental Information 1 Raw data

Click here for additional data file.

Supplemental Information 2 Flow cytometry data

Click here for additional data file.

Supplemental Information 3 Author checklist - full

Click here for additional data file.

Supplemental Information 4 Figs. S1, S2, and S3 raw data

Click here for additional data file.

Supplemental Information 5 Effects of different concentrations of colivelin TFA on cell viability

Click here for additional data file.

Supplemental Information 6 Effects of different concentrations of Ac2-26 on cell viability

Click here for additional data file.

Supplemental Information 7 Effects of different concentrations of Ac2-26 on ALT activity in serum

Click here for additional data file.

Supplemental Information 8 Ratio of NADP + to NADPH

Click here for additional data file.

Additional Information and Declarations

Competing Interests

Author Contributions

Animal Ethics

Data Availability

The authors declare there are no competing interests.

Wanzhen Li conceived and designed the experiments, performed the experiments, analyzed the data, prepared figures and/or tables, and approved the final draft.

Hongxin Jiang conceived and designed the experiments, prepared figures and/or tables, authored or reviewed drafts of the article, and approved the final draft.

Chen Bai conceived and designed the experiments, performed the experiments, prepared figures and/or tables, and approved the final draft.

Shuna Yu conceived and designed the experiments, prepared figures and/or tables, authored or reviewed drafts of the article, and approved the final draft.

Yitong Pan conceived and designed the experiments, performed the experiments, prepared figures and/or tables, and approved the final draft.

Chenchen Wang performed the experiments, prepared figures and/or tables, and approved the final draft.

Huiting Li performed the experiments, prepared figures and/or tables, and approved the final draft.

Ming Li performed the experiments, prepared figures and/or tables, and approved the final draft.

Yaxin Sheng performed the experiments, prepared figures and/or tables, and approved the final draft.

Fangfang Chu analyzed the data, prepared figures and/or tables, and approved the final draft.

Jie Wang analyzed the data, prepared figures and/or tables, and approved the final draft.

Yuting Chen analyzed the data, prepared figures and/or tables, and approved the final draft.

Jianguo Li conceived and designed the experiments, authored or reviewed drafts of the article, and approved the final draft.

Jiying Jiang conceived and designed the experiments, authored or reviewed drafts of the article, and approved the final draft.

The following information was supplied relating to ethical approvals (i.e., approving body and any reference numbers):

Weifang Medical University Committee for Laboratory Animal Research (the Animal Ethics Committee which is responsible for ethical evaluation and surveillance of animal studies) provided full approval for this research (No. 2020SDL187)

The following information was supplied regarding data availability:

The data is available at Figshare: Jiang, Jiying (2022): Supplemental files.zip. figshare. Journal contribution. https://doi.org/10.6084/m9.figshare.19802902.v1.

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
