# Peer review of "Ac2-26 attenuates hepatic ischemia-reperfusion injury in mice via regulating IL-22/IL-22R1/STAT3 signaling"

_PeerJ, doi:10.7717/peerj.14086_

## Round 0.1 · original submission · Major Revisions

Accurate revision of the material and methods section after adding all missed information is highly recommended.

Reviewer 1 ·

Basic reporting

Regarding the manuscript entitled “Ac2-26 attenuates hepatic ischemia-reperfusion injury via regulating IL-22/IL-22R1/STAT3 signaling”, the authors investigated the in vivo and in vitro effects of Ac2-26, a short peptide of Annexin A1 protein, against hepatic ischemia-reperfusion injury via suppression of oxidative stress and apoptosis. They assumed that Ac2-26 could protect the ischemic/reperfused liver by inhibiting the IL-22/IL22R1/STAT3 signaling pathway. Indeed, the study results are merit and potentially interesting. However, several points are raised.
Comments
1- Abbreviations should be revised throughout the manuscript.
2- Figure legends should be revised.
3- Histological micrographs should be labeled by stars, arrows, stricks, ….. to identify the histological findings.
4- Lack of references in the methods section. Each method should be referenced with relevant reference(s)
5- “1% pentobarbital sodium”, explain why the authors used this method of anesthesia during the experiment. However, pentobarbital causes hypotension, bradycardia, and decreased respiratory rate.
6- What about the mortality rate in this experiment?
7- Suzuki's score should be explained in more detail. Please add references
8- Quantitative real-time PCR (qRT–PCR) section, a more detailed methodology is required.
9- The results do not make sense. It is better to rewrite the result section with a more comparable pattern using fold change or percent expressions.
10- The quality of the supplied western blot bands is very poor. They are not like strips separated from the same gel. Provide original strips when necessary
11- Measurement of direct oxidative stress markers like H2O2, HO-, NADPH oxidase, …. etc may add benefits to the study, proving

Experimental design

Good experimental design

Validity of the findings

no comment

Reviewer 2 ·

Basic reporting

No comment

Experimental design

No comment

Validity of the findings

No comment

Additional comments

I found that this paper to be overall very well informed, objective and well written, however certain points need to be clarified before the manuscript could be considered for publication.
1. Please give specific category numbers of all chemicals and kits used in this study so people can cite and repeat the work in the future if wanted.
2. Please give the name of the food if it was commercial or the ingredients if it was made in your lab.
3. Please give centrifugation speed in g instead of rpm.
4. All p in the manuscript should be capital and italics.
5. Amount of liver used for RNA extraction should be mentioned.
6. The references of histopathological examination should be provided.
Minor points
L204 Anzyme-linked immunosorbent assay (ELISA) analysis should be replaced by Enzyme-linked immunosorbent assay (ELISA) analysis

---

## Round 0.2 · accepted · Accept

The authors have addressed all of the reviewers' comments.

Reviewer 2 ·

Basic reporting

no comment

Experimental design

no comment

Validity of the findings

no comment